# Single Cell Sequencing Reveals Mechanisms of Persistent Truncus Arteriosus Formation after *PDGFRα* and *PDGFRβ* Double Knockout in Cardiac Neural Crest Cells

**DOI:** 10.3390/genes13101708

**Published:** 2022-09-23

**Authors:** Tianyun Chen, Shen Song, Haobin Jiang, Hong Lian, Shengshou Hu

**Affiliations:** 1State Key Laboratory of Cardiovascular Disease, Fuwai Hospital, National Center for Cardiovascular Disease, Chinese Academy of Medical Sciences and Peking Union Medical College, Beijing 100006, China; 2Division of Thoracic Surgery, School of Medicine, First Affiliated Hospital, Zhejiang University, Hangzhou 310027, China

**Keywords:** single-cell RNA-seq, cardiac outflow tract, persistent truncus arteriosus, neural crest cells, platelet-derived growth factor receptor

## Abstract

Persistent truncus arteriosus (PTA) is an uncommon and complex congenital cardiac malformation accounting for about 1.2% of all congenital heart diseases (CHDs), which is caused by a deficiency in the embryonic heart outflow tract’s (OFT) septation and remodeling. *PDGFRα* and *PDGFRβ* double knockout (DKO) in cardiac neural crest cells (CNCCs) has been reported to cause PTA, but the underlying mechanisms remain unclear. Here, we constructed a PTA mouse model with *PDGFRα* and *PDGFRβ* double knockout in *Pax3*^+^ CNCCs and described the condensation failure into OFT septum of CNCC-derived cells due to disturbance of cell polarity in the DKO group. In addition, we further explored the mechanism with single-cell RNA sequencing. We found that two main cell differentiation trajectories into vascular smooth muscle cells (VSMCs) from cardiomyocytes (CMs) and mesenchymal cells (MSs), respectively, were interrupted in the DKO group. The process of CM differentiation into VSMC stagnated in a transitional CM I-like state, which contributed to the failure of OFT remodeling and muscular septum formation. On the other hand, a *Penk*^+^ transitional MS II cluster closely related to cell condensation into the OFT septum disappeared, which led to the OFT’s septation absence directly. In conclusion, the disturbance of CNCC-derived cells caused by *PDGFRα* and *PDGFRβ* knockout can lead to the OFT septation disorder and the occurrence of PTA.

## 1. Introduction

Congenital heart diseases (CHDs) are the most common congenital disabilities, mainly resulting from the disruption of discrete subsets of cardiac progenitor cells [1]. Persistent truncus arteriosus (PTA) is a rare but complex congenital cardiac malformation accounting for about 1.2% of all CHDs [2]. About 80% or more of children born with PTA die within the first year of life without surgical intervention [2,3,4,5]. PTA is marked by a single great artery arising from the base of the heart, with a ventricular septal defect (VSD), leading to the mixing of systemic, coronary, and pulmonary blood flow. These traits are believed to result from the failure of proper separation of the cardiac outflow tract (OFT).

Neural crest cells (NCCs) are a group of multipotent, migratory cells that delaminate from the neural tube’s dorsal part and contribute to the development of various tissues and organs [6,7,8]. The subpopulation of NCCs contributing to the heart is called cardiac neural crest cells (CNCCs) [9,10]. These cells were firstly reported to play important roles in cardiovascular development, including OFT septation, which ensures the connection between the embryonic heart chambers and vascular network, by Le Lievre et al. [11]. Initially, a solitary tube called the truncus arteriosus was formed, then gradually remodeled into two tubes, giving rise to the aorta and pulmonary arteries [12] (Figure 1A). This remodeling is one of the most sensitive processes in heart morphogenesis. For instance, faulty septation of the OFT accounts for 30% of all CHDs. This prompts us to better understand the cellular and molecular cues that CNCCs participate in the OFT separation during development [13].

CNCCs require multiple environmental signals, including wingless/integrated, bone morphogenetic proteins, and fibroblast growth factors to be specified, migrate, proliferate, differentiate, and survive when they leave the neural tube and migrate into the pharynx and the OFT [14,15,16]. Combined with the above signaling cues, members of the *ephrin* family of ligands and receptors, semaphorins (including receptors plexin-A2, plexin-D1, and *Nrp1* and its ligands *Sema6a*, *Sema6b*, and *Sema3c*) and connexin 43, define specific routes for the migration of CNCCs into the OFT [17,18]. These signal pathways and genes are closely related to the regulation of CNCCs in the normal development of the OFT.

Platelet-derived growth factor receptor (*PDGFR*) is a surface tyrosine kinase receptor of the PDGF family. It is involved in regulating cell chemotaxis, proliferation, and differentiation. In addition, it plays an essential role in regulating fetal growth and development, wound repair, and tumorigenesis [19]. *PDGFR* has two subforms: *PDGFRα* and *PDGFRβ* [20]. As reported by Richarte et al. [21], embryos with *PDGFRα* conditional knockout in *Wnt1*^+^ NCCs developed PTA (penetrance 25%) and VSD (penetrance 75%), while those with *PDGFRβ* conditional knockout showed no obvious abnormity. Simultaneous knockout of *PDGFRα* and *PDGFRβ* in *Wnt1*^+^ CNCCs could induce PTA with 100% penetrance, suggesting that *PDGFRα* and *PDGFRβ* may modulate the involvement of *Wnt1*^+^ CNCCs in the OFT separation process. However, despite exploring the phenotype and cellular mechanisms of *PDGFRα* and *PDGFRβ* double knockout (DKO), the study did not include further mechanistic investigation at the molecular level after *PDGFRα* and *PDGFRβ* DKO [21]. This research explored the changes after *PDGFRα* and *PDGFRβ* DKO in CNCCs. Given that gene expression might provide information essential to understanding PTA, we first constructed a PTA embryo mouse model with *PDGFRα* and *PDGFRβ* simultaneous knockout in *Pax3*^+^ CNCCs and elucidated the pathological consequences. Importantly, we performed single-cell RNA-sequencing of OFT cells to systematically elucidate the key cell behaviors and molecular mechanisms in this process. Our results provided an OFT cell profile at the septation stage, as well as cytological and molecular changes after *PDGFRα* and *PDGFRβ* knockout in CNCCs, highlighting that the disturbance of cell differentiation led to the failure of septal-bridge construction as a possible mechanism of PTA malformation.

## 2. Materials and Methods

### 2.1. Animal Models

Embryonic 11.5-, 12.0-, 12.5-, 14.5-, and 17.5-day-old, and neonatal 1-day-old mice of both sexes were used in our study. The C57BL/6J WT mice (No. 213) were obtained from Vital River Laboratory Animal Technology Co. Ltd. (Beijing, China). The *Pax3*^cre/+^ (B6;129-*Pax3^tm1(cre)Joe^*/J, JAX 005549), *PDGFRα^fl/fl^* (B6.Cg-*PDGFRa^tm8Sor^*/EiJ, JAX 006492), *PDGFRβ^fl/fl^* (129S/SvJae-*PDGFRb^tm11Sor^*/J, JAX 010977) and *Rosa26^tdTomato^* (B6;129S6-*Gt(ROSA)26Sor^tm14(CAG-tdTomato)Hze^*/J, JAX 007908) mice were acquired from the Jackson Laboratory (Bar Harbor, ME, USA). We first obtained *Pax3*^cre/+^; *PDGFRα^fl/+^*; *PDGFRβ^fl/fl^* and *PDGFRα^fl/fl^*; *PDGFRβ^fl/fl^* as the parental generation through hybridization and genotype identification. Then, we constructed *PDGFRα* and *PDGFRβ*-conditional-double-knockout mice from the hybridization of *Pax3^cre/+^; PDGFRα^fl/+^; PDGFRβ^fl/fl^* and *PDGFRα^fl/fl^; PDGFRβ^fl/fl^* parental generation mice. The littermates were used as the control group. Embryos with PDGFRα conditional knockout developed various cardiac malformations, including PTA (penetrance 25%) and VSD (penetrance 75%), while simultaneous DKO of *PDGFRα* and *PDGFRβ* (DKO) could induce PTA with 100% penetrance. Our research aims to explore the mechanism of PTA based on stably constructing the mouse model. Thus, transcriptomes of single-cell RNA sequencing were captured from control and DKO group embryos, as described below. Lineage tracing of CNCC-derived cells was performed using *Pax3^cre/+^; PDGFRα^fl/fl^; PDGFRβ^fl/fl^; Rosa26^tdTomato^* genotype mice, obtained from the hybridization of *Rosa26^tdTomato^;*
*Pax3*^cre/+^; *PDGFRα*^fl/+^*; PDGFRβ^fl/fl^* and *Rosa26^tdTomato^;*
*PDGFRα^fl/fl^; PDGFRβ^fl/fl^* mice. All experiments involving animals were conducted in accordance with the Guide for the Use and Care of Laboratory Animals [22]. All animal protocols were approved by the Institutional Animal Care and Use Committee (IACUC), Fuwai Hospital, Chinese Academy of Medical Sciences (FW-2021-0044).

In determining the stage of embryonic development, noon on the day of plug detection after sexually mature male and female (6–8 weeks of age) mice were caged together at a ratio of 3:1 was considered E0.5. Pregnant females were euthanized to harvest embryos at E12.5 for single-cell RNA sequencing and section-based immunofluorescence experiments, at E11.5, 12.0, 12.5, 14.5, and 17.5 for confirmation of cardiac phenotype, and at P0 for statistics of birth phenotypes. Phenotyping criteria at E17.5: (1) PTA + VSD-aorticopulmonary septum is absent and replaced by the truncus arteriosus. All PTA were combined with VSD in our models; (2) DORV-aorticopulmonary septum exists, but both aorta and pulmonary artery originate from the right ventricle, with or without VSD, respectively defined as DORV + VSD and isolated DORV; (3) isolate VSD-ventricular septum defect exists, without any aorta or pulmonary artery defect; (4) Normal-no obvious cardiac structural abnormality exists. The sample size of embryos used for single-cell transcriptome analysis at E12.5 was selected to obtain cell numbers equivalent to the estimated cell numbers in the OFT region.

### 2.2. Embryo Dissection and Single-Cell Library Generation

The entire OFT region was micro-dissected at E12.5. Embryos were dissected in cold PBS and placed in PBS with 1% FBS (Gibco, 10437010, Life Technologies, Carlsbad, CA, USA) solution on ice until dissociation. Embryo DNA was extracted and used for genotyping to distinguish the control and DKO embryos before further microdissection of OFT regions.

Dissected OFT tissue was dissociated into single-cell suspension with Pierce Primary Cardiomyocyte Isolation Kit (ThermoFisher Scientific, Waltham, MA, USA, 88281). According to protocol, OFT tissue was first rinsed with HBSS (Gibco, 14025076, Life Technologies, Carlsbad, CA, USA) for 5 min. Then each OFT was added to the 1.5 mL centrifuge tube with preset enzyme solution (20 μL recombinant isolation enzyme 1 with papain and 1 μL isolation enzyme 2 with thermolysin) and digested for 30 min in the 37 °C water bath. 150 μL DMEM (Gibco, 10313021, Life Technologies, Carlsbad, CA, USA) containing 10% HFBS (Gibco, 10437085, Life Technologies, Carlsbad, CA, USA) was added to each tube, and the cells were fully dissociated by blowing 25–30 times with sterile straw manually instead of using the gentleMACS Dissociator. The cell suspension was centrifuged at 300× *g* for 5 min. After discarding the supernatant, cells were resuspended in 100 μL HBSS with 0.04% BSA (Gibco, GP22900, Life Technologies, Carlsbad, CA, USA). Cell viability, aggregation rate, and concentration were measured by Countstar (ABER, Aberystwyth, UK) to ensure sample quality.

Single-cell libraries from this suspension were generated according to the manufacturer’s instructions in the Chromium Single Cell 3′ Reagent Kit (10X Genomics, Pleasanton, CA, USA) v.3 User Guide in the 10X Genomics Chromium controller. The cell capture efficiency of the Chromium controller is ~50%. Thus, we loaded all cells dissected from embryos. Additional components used for library preparation include the Chromium Single Cell 3′ Library and Gel Bead Kit (10X Genomics, Pleasanton, CA, USA) v.3 and the Chromium Single Cell 3′ Chip kit (10X Genomics, Pleasanton, CA, USA) v.3. Our sequencing data have been submitted to Sequence Read Archive (SRA). The bioproject number is PRJNA803335, and the submission number is SUB11005893.

### 2.3. Single-Cell Transcriptome Library Preparation and Sequencing

Libraries were prepared according to the manufacturer’s instructions using the Chromium Single Cell 3′ Library and Gel Bead Kit (10X Genomics, Pleasanton, CA, USA) v.3 and Chromium i7 Multiplex Kit (10X Genomics, Pleasanton, CA, USA). Final libraries were sequenced on an Illumina NovaSeq 6000 System (Illumina, San Diego, CA, USA). Developmental-timing-matched control and *PDGFRα* and *PDGFRβ*-conditional-knockout libraries from each litter were pooled and sequenced in the same lane. Sequencing parameters were selected according to the Chromium Single Cell (10X Genomics, Pleasanton, CA, USA) v.3 specifications. Four libraries were sequenced to a mean read depth of at least 50,000 aligned reads per cell.

### 2.4. Processing of Raw Sequencing Reads

Raw sequencing reads were processed using the Cell Ranger (10X Genomics, Pleasanton, CA, USA) v.4.4.0 pipeline from 10X Genomics (https://support.10xgenomics.com, accessed on 25 October 2020). In brief, raw base call files generated by Illumina sequencers were demultiplexed into reads in FASTQ format using the “cellranger mkfastq” pipeline. Then, the reads of each library were processed separately using the “cellranger count” pipeline to generate a gene-barcode matrix for each library. During this step, the reads were aligned to the mouse (*Mus musculus*) reference genome (version: mm10). Cell barcodes and UMIs associated with the aligned reads were subjected to correction and filtering. The resulting gene-cell UMI count matrices for each sample (control and DKO, E12.5) were then concatenated into 1 matrix using the “cellranger aggr” pipeline, which also normalized the libraries to the same sequencing depth.

### 2.5. Cell Filtering and Cell-Type Clustering Analysis

We sequenced the transcriptomes of 7506 cells captured from control and 8529 cells captured from DKO embryos in total. Further filtering and clustering analyses of these cells were performed with the Seurat v.3.6 R package, as described in the tutorials (http://satijalab.org/seurat/) [23]. For each aggregated dataset (control and DKO), cells were normalized for genes expressed per cell and total expression, then multiplied by a scale factor of 10,000 and log-transformed. Cells that were of low quality or represented doublets were excluded from our analyses—this was achieved by filtering out cells with (1) greater than 25,000 and fewer than 600 genes; (2) greater than 4000 and fewer than 350 UMI counts; (3) greater than 20% mitochondrial gene proportion; and (4) greater than or equal to 40% ribosomal gene proportion using Seurat (Appendix A). We then performed a generalized linear regression on all genes to eliminate technical variability due to the number of genes detected, the number of UMIs, and the cell cycle stage (ScaleData function). Two thousand highly variable genes in the dataset were computed and used as input for principal component analysis. Significant principal components were used for downstream graph-based clustering, and uniform manifold approximation and projection (UMAP) [24] dimensionality reduction was used to project these populations in 2 dimensions. For clustering, the resolution parameter (resolution = 0.5), which indirectly controls the number of clusters, was approximated based on the number of cells according to Seurat guidelines; a vector of resolution parameters was passed to the FindClusters function, and the optimal resolution that established discernible clusters with distinct marker gene expression was selected. To identify marker genes, the clusters were compared pairwise for differential gene expression using the Wilcoxon rank-sum test for single-cell gene expression (FindAllMarkers function, min.pct = 0.25, min.diff.pct = 0.25, return.thresh (adjusted *p*-value cut-off) = 0.05). To detect differentially expressed genes between DKO and control groups, we conducted Wilcoxon rank-sum test implemented in “FindMarkers” function (min.diff.pct = 0.1, return.thresh (adjusted *p*-value cut-off) = 0.05). To conduct gene function enrichment analysis, we chose Metascape biological process database as the resource on gene functions. Enrichment analyses were carried out with an adjusted *p*-value threshold of 0.05.

### 2.6. RNA Velocity Analysis

We performed RNA velocity analysis using scVelo (version 0.2.2) [25] for inferring cell trajectories. The combined loom file was normalized and log transformed with function scvelo.pp.filter_and_normalize. The first and second order moments for each cell across its nearest neighbors were computed with function scvelo.pp.moments (n_neighbors = 30, n_pcs = 30). Then, the velocities were estimated by running the likelihood-based dynamical model with function scvelo.tl.velocity (mode = “deterministic”), and the velocity graph was constructed with scvelo.tl.velocity_graph. RNA velocities were then recomputed with functions scvelo.pp.neighbor, scvelo.tl.velocity and scvelo.tl.velocity_graph. To visualize the RNA velocities, the UMAP coordinate information and cluster assignments were extracted from the Seurat analysis output. Velocities were projected into 2D on top of the previously computed UMAP coordinates.

### 2.7. CellChat Analysis

To study the communicating interactions between cells and identify the mechanism of the communicating molecules at a single-cell resolution, R package “CellChat” (version 1.0.0) was applied to cells involved in 6 cell clusters, including MSI, CM I, VSMC I, MS II, CM II and VSMC II [26].

### 2.8. Immunofluorescence Experiments

Each immunofluorescence experiment was replicated at least twice for identifying spatial expression of genes and 3 times for quantification of in situ signal for differentially expressed genes in the *PDGFRα* and *PDGFRβ*-conditional-knockout analysis. For immunofluorescence experiments performed on E12.5 heart sections: Embryos were washed 5 times in PBS after overnight fixation in 4% formaldehyde and stored in 75% ethanol indefinitely until embedding. Embryos were embedded in paraffin processed using standard protocols and embedded for transverse sectioning. Tissue slices were serially sectioned at 5 μm intervals, mounted on slides, and stored at room temperature until the immunofluorescence experiment for paraffin-embedded sections. Sections were imaged with a Zeiss Axio Observer Z1 inverted epifluorescence microscope (Carl Zeiss, Jena, Germany) with Zeiss Axiocam MRm and PCO.edge sCMOS (PCO Imaging) monochrome cameras run by Zeiss Zen imaging software (Carl Zeiss, Jena, Germany). Catalog numbers (Abcam, Cambridge, UK) for Immunofluorescence antibodies used in this study: Anti-*PDGFR* α: ab234965; Anti-*PDGFR* beita: ab69506; Anti-α smooth muscle Actin: ab8211; Anti-GM130: ab52649.

### 2.9. In Situ Hybridization Experiments

Each immunofluorescence experiment was replicated at least twice for identifying spatial expression of genes and 3 times for quantification of in situ signal for differentially expressed genes in the *PDGFRα* and *PDGFRβ*-conditional-knockout analysis. For in situ hybridization experiments performed on E12.5 heart sections: hearts were washed 5X in PBS after overnight fixation in 4% formaldehyde and stored in 70% ethanol (VWR, Arlington Heights, IL, USA, 89125–186) indefinitely until embedding. Embryos were embedded in Histogel (Thermo Scientific, Waltham, MA, USA, R904012) and paraffin processed using standard protocols and embedded for transverse sectioning. Tissue slices were serially sectioned at 5 μm intervals, mounted on slides, and stored at room temperature until initiation of the RNA Scope (ACD, Santa Ana, CA, USA) protocol for paraffin-embedded sections (User manual catalog number 322452-USM). Sections were imaged with a Zeiss Axio Observer (Carl Zeiss, Jena, Germany). Z1 inverted epifluorescence microscope (Carl Zeiss, Jena, Germany) with Zeiss Axiocam MRm and PCO edges CMOS (PCO. Imaging, Kelheim, Germany) monochrome cameras run by Zeiss Zen imaging software (Carl Zeiss, Jena, Germany). The catalog numbers for RNA Scope probes used in this study were: *Penk*, 318761-C2; *Sema3c*, 441441.

### 2.10. Statistics and Reproducibility

Standard statistical analyses were performed using GraphPad Prism 8. The number of replicates, statistical test used, and test results were described in the figure legends. For fluorescence quantification, corrected total fluorescence values were log-transformed before t-tests were conducted to satisfy the prerequisite assumptions of normality. For all quantifications, *p* values were calculated using 2-tailed tests. The mean ± s.e.m. was reported. The level of significance in all graphs is represented as follow: * *p* < 0.05 and ** *p* < 0.01. When representative results were presented to indicate expression patterns of genes in wild-type embryos, at least 2 independent embryos were analyzed. Differential proportion analysis (DPA, developed by Farbehi et al.) was used to analyze whether changes in the proportion of populations were greater than expected by chance [27].

## 3. Results

### 3.1. Simultaneous Knockout of PDGFRα and PDGFRβ in Pax3^+^ NCCs Induced PTA

To ablate *PDGFRα* and *PDGFRβ* in CNCCs, we first constructed a *Pax3^cre/+^; PDGFRα^fl/fl^; PDGFRβ^fl/fl^* transgenic mouse strain (Figure 1C) [10]. Cell lineage tracing of CNCC-derived cells was achieved using a ubiquitous fluorescent Cre reporter allele, *Rosa26^tdTomato^*, in which Cre-mediated recombination labels cells with a membrane-targeted red fluorescent protein (RFP) [28]. The pattern of cell recombination in the OFT cushions of E12.5 control embryos carrying either Cre driver matched with the pattern of colonizing CNCC described by previous lineage analyses [29]. In situ hybridization on histological sections was used to monitor *PDGFRα* and *PDGFRβ* expression and validate their deletion efficiency on CNCCs. As shown in Figure 1D, *PDGFRα* and *PDGFRβ* were ubiquitously expressed in all layers of control embryos. In *Pax3^cre/+^; PDGFRα^fl/fl^; PDGFRβ^fl/fl^; Rosa26^tdTomato^* recombined embryos, the CNCC-derived cells displayed a substantial reduction in *PDGFRα* and *PDGFRβ* levels, while the surrounding tissues remained *PDGFRα* and *PDGFRβ* positive (Figure 1D). *PDGFRα* and *PDGFRβ* play an important role in the spatial distribution and function of CNCC-derived cells.

Genotype identification of surviving embryos at E17.5 showed that genes were inherited according to the Mendelian law (Appendix A). Embryonic lethality was observed in mice with simultaneous knockout of *PDGFRα* and *PDGFRβ* in *Pax3^+^* CNCCs at P0, but not in single knockouts (Appendix A). Coronal sections of E17.5 embryo mouse hearts were harvested to identify the cause of embryonic lethality. The phenotypic confirmation at E17.5 with multiple methods (stereoscope, microscope, and 3D OFT reconstruction) all showed that loss of *PDGFRα* and *PDGFRβ* in *Pax3^+^* CNCCs resulted in PTA with 100% penetrance (Figure 1E), consistent with a previous report [21]. We also performed phenotype analysis in mice with single-gene knockout of *PDGFRα* (*Pax3^Cre/^*, *PDGFRα^fl/fl^*) or *PDGFRβ* (*Pax3^Cre/+^; PDGFRβ**^fl/fl^*)*. PDGFRα* knockout mice carried a series of conotruncal malformations, including PTAs (31%), the double outlet of the right ventricle (DORV, 53%), and isolated VSDs 8% (Figure 1F and Appendix A). In comparison, most *PDGFRβ* mice remained normal except for a few isolated VSDs (5%) (Figure 1F and Appendix A). The phenotypes of single and double knockout groups indicated that conditional *PDGFRα* or *PDGFRβ* DKO in CNCCs could lead to conotruncal malformations and have a functional compensation between *PDGFRα* and *PDGFRβ* (Appendix A). In addition, we observed that the conotruncal phenotypes of *Pax3^Cre/+^; PDGFRα^fl/+^; PDGFRβ^+/+^* embryos were normal (Appendix A). This indicated that the specific knockdown of *PDGFRα* in CNCCs did not affect the development of the OFT.

Since the phenotype of PTA is highly correlated with abnormality in OFT separation, we checked OFT separation of embryos at different stages (E11.5, E12.0, E12.5, and E14.5) after simultaneous knockout of *PDGFRα* and *PDGFRβ* in CNCCs. To describe the morphological progression of OFT separation, we analyzed 65 hearts by hematoxylin-eosin (HE) staining (E11.5: 16; E12.0: 18; E12.5: 21; and E14.5: 10). Horizontal OFT sections showed that the *PDGFRα* and *PDGFRβ* DKO and control group embryos were normal, and no gross morphological defects were detected at E11.5 and E12.0. At E12.5, DKO embryos showed penetrant morphological OFT defects, and the cushion was poorly matched and failed to form the septal bridge structure. At E14.5, OFT separation between the aorta and pulmonary artery was completely absent in DKO mice (Figure 1G). Phenotypic examination of transverse sections at three distinct distal-proximal levels of OFT (distal, medial, and proximal) also showed that the lack of septal bridge structure in the medial level led to the separation failure in the distal level directly (Figure 1H). This indicated that *PDGFRα* and *PDGFRβ* combined knockout in CNCCs could lead to disorder of septal bridge formation and absence of OFT separation, which may trigger the induction of the PTA phenotype. In addition, time course phenotypic changes of OFT were observed for the first time from E12.5 and gradually developed into PTA from E12.5 (Figure 1G). Thus, it is a crucial time point for studying this process’s cellular and molecular mechanisms, which are suggested by the results of early phenotypic confirmation.

### 3.2. Condensation Failure of CNCC-Derived Cells Due to Disturbance of Cell Polarity after Pdgfrα and Pdgfrβ Knockout

Next, we next investigated the cellular mechanisms, including cell migration, proliferation, apoptosis, and polarity in OFT separation after *PDGFRα* and *PDGFRβ* DKO. First, whole-mount immunostaining in CNCC lineage-tracing mice revealed that RFP^+^ CNCCs reached similar OFT levels in E12.5, indicating that cell migration was not affected by *PDGFRα* and *PDGFRβ* DKO (Appendix A). Next, we analyzed the proliferation of cells within the conotruncal region of embryos at E12.5 to assess the proliferation rate of CNCC-derived cells in *PDGFRα* and *PDGFRβ* DKO and control embryos. Quantifying cell proliferation in tissue sections, using antibodies against cell proliferation-related markers Ki67 and phospho-histone H3 (pH3) demonstrated no significant change in the number of RFP^+^ CNCC-derived proliferating cells compared to littermate controls (Appendix A). Terminal deoxynucleotidyl transferase-mediated dUTP-biotin nick end labeling (TUNEL) assay revealed similar apoptosis rates in both groups at E12.5 (Appendix A). These results indicated that *PDGFRα* and *PDGFRβ* do not control the proliferation or survival of CNCCs. Finally, we wondered whether OFT morphogenetic defects in the DKO group resulted from differences in cell polarity because cell polarity may reflect the cell condensation in OFT cushions, an important segment in septum formation [13]. We performed Golgi marker GM130 and nuclear DAPI immunostaining, determined the polarity direction of OFT cells through the connecting line between Golgi and nuclear center (taking the craniocaudal axis of mice as 0°), and statistically analyzed the position and orientation of all cells at the same level (Figure 2A). In the control group, the polarity of OFT cushion cells on the left side was more concentrated between 120° to 150°, which was consistent with the fusion angle of OFT cushions. However, the cellular polarity was not apparent in the DKO group, suggesting that OFT cell polarity was potentially disrupted in DKO embryos (Figure 2B and Appendix A).

In summary, the above results showed that *PDGFRα* and *PDGFRβ* are important modulators of CNCC behavior in the heart and the OFT separation process. After *PDGFRα* and *PDGFRβ* DKO, the polarity of OFT cells is affected. Without correct cell polarity concentration in the fusion angle of OFT cushions, loosely arranged cells in the middle of OFT may fail to condense into a compact septal-bridge structure, leading to the septum formation disorder and the emergence of the PTA phenotype.

### 3.3. Single-Cell Transcriptomic Sequencing and Overview of Developing OFT Cells

Although traditional gene knockout studies have revealed that *PDGFRα* and *PDGFRβ* can modulate the involvement of *Pax3*^+^ CNCCs in the OFT separation process, but the specific mechanisms of gene regulation are still elusive. Given the complexity of the OFT separation process, we used single-cell transcriptomic sequencing to further its exploration.

We isolated OFT cells from embryonic mouse hearts at the OFT separation stage (E12.5). Then, we conduct single-cell transcriptomic sequencing. After stringent quality filtering (details see Methods), we obtained a high-quality transcriptomic dataset of 6337 cells in the DKO group and 5661 cells in the control group (Figure 3A, Appendix A). Unsupervised clustering identified 10 cell clusters representing distinct cell types or subpopulations (Figure 3B and Appendix A).

We performed differential expression analysis between each cluster and all others to identify each cell cluster and assigned a specific cell type to each cluster based on established lineage-specific marker genes (Figure 3C and Appendix A). Epicardial cells (EP) specifically expressed *Upk3b* and *Upk1b19* [30]. Endocardial cells (EC) I and II clusters highly expressed endocardial markers *Ramp2* and *Fabp5* [31]. The cardiomyocyte (CM) lineage constituted CM cluster I, II, and III, as they highly expressed CM marker genes, such as *Myh7* and *Myl4* [32,33]. The vascular smooth muscle cell (VSMC) lineage comprised two closely aligned clusters, namely, VSMC I and II, which highly expressed *Cxcl12*, a chemokine encoding gene that is highly expressed in the walls of the aorta and pulmonary trunk of the embryonic heart [34] and *Rgs5*, a gene that is abundantly expressed in arterial smooth muscle cells [35,36]. The mesenchymal lineage included mesenchymal cells (MS) I and II clusters, which specifically expressed mesenchymal marker genes such as *Postn* and *Cthrc1* (Figure 3D) [32].

Significantly, we found significant changes in the cell type proportions following DKO, mainly in CMs and MSs (*p <* 0.05). In CMs, the proportion of CM I cluster increased, and CM II decreased. Meanwhile, the MS II cluster was absent in the DKO group (Figure 3E).

### 3.4. The Disturbance of CMs Trans-Differentiation after Pdgfrα and Pdgfrβ Knockout

We bioinformatically isolated three CM clusters from the rest of the cells in the OFT for further analysis (Figure 4A). After *PDGFRα* and *PDGFRβ* knockout in CNCCs, the proportion of CM I cluster increased, and CM II decreased significantly (*p <* 0.05), while CM III remained similar (Figure 3E and Figure 4B). Differentially expressed genes (DEGs) analysis showed that the top 5 marker genes in CM I were *Malat1*, *Itm2a*, *Actb*, *Bmp4*, and *Clu*; *Mest*, *Gja1*, *mt-Atp6*, *Atp2a2*, and *Nppb* in CM II; and *Cited1*, *Myl2*, *Smpx*, *Hopx*, and *Gyg1* in CM III (Appendix A). Pathway enrichment analysis pathway for each CM cluster showed that the genes up-regulated in CM II and III were enriched for oxidative phosphorylation and CM contraction, with high expression of *Smpx*, *Nppb*, and *Myl2* (Figure 4C). By contrast, in CM I, terms was mainly enriched in vascular development, ECM organization, and cell migration, with high expression of *Cxcl12* (Figure 4C).

The highly expressed genes in CM I included *Cox4i2*, *Col1a2*, *Cxcl12*, and *Eln*, which were specifically expressed in VSMCs [37]. In addition, previous studies have found that *Sema3c* was expressed in CMs of the OFT wall and played a vital role in attracting CNCCs into OFT [9,17,18,38] and promoting the aggregation of CNCCs in primary cultures [17,18,39,40]. It was also reported that *Sema3c^+^* CMs were mainly distributed on the pulmonary artery (Pa) side and represented future subpulmonary CMs derived from the second heart field [17,38]. In our data, *Sema3c* was highly expressed in the CM I cluster, consistent with previous findings (detail see Result 5, Figure 5G). In addition, the *Sema3c*-*Nrp1* signaling has been proposed as a triggering signal for the epithelial-mesenchymal transition (EMT) of the endocardium [39]. Consistent with this, highly expressed marker genes, including *Malat1*, *Bmp4*, *Clu*, *Cxcl12*, *Col1a2,* and *Mt1,* were closely related to the EMT procedure [40,41,42,43,44,45]. These results indicated that the CM I cluster might represent future subpulmonary CMs and had VSMC-like characteristics (*Cxcl12^+^*), which also played an important role in EMT regulation. CM II cluster highly expressed mature CM markers like *Nppb* and electrophysiology genes like *Atp2a2* represented a relatively mature working CM cluster [46]. While CM III highly expressed *Hopx*, which contributes to stem cell quiescence, might represent multipotent cardiomyoblasts [47] (Figure 4C,D).

Compared with the control group, up-regulated genes in the DKO group included *Hspa1a*, *Hspa1b,* and *Malat1* (Figure 4E and Appendix A). Pathway enrichment analysis revealed that up-regulated genes participated in L13a-mediated translational silencing of ceruloplasmin expression and regulation of cellular protein catabolic process (Figure 4F). In CM II, down-regulated genes included *Cited1*, *Fos*, *Tnni1*, *Smpx*, and *Egr1*, whereas up-regulated genes included *Mt1*, *Malat1*, *Hk2*, *Bnip3*, and *Ttn*. In CM III, down-regulated genes included *Myl1*, *Cited1*, *Atox1*, *Mgst3*, and *Tnni1*, while up-regulated genes included *Bsg*, *Nme2*, *Hk2*, *Rpl13a*, and *Ankrd1* (Figure 4E). Enrichment pathways in CM II included ATP metabolic process and cardiac muscle tissue development, while CM III included ATP metabolic process and GTP hydrolysis (Figure 4F). RNA velocity analysis is a new analytical method that can infer the cell differentiation trend based on the relative abundance of spliced and unspliced transcripts [48]. Our RNA velocity analysis result of CMs (Figure 4G) and all OFT cells (Appendix A) revealed that CM II showed progression toward VSMC via CM I. In addition, genes related to vascular smooth muscle development, including *Csrp2* and *Acta2,* were down-regulated in CM I in the DKO group (Figure 4E) and mainly participated in vascular development, energy metabolism, and smooth muscle contraction pathways, which were similar to CM I marker genes (overlapped gene including *Csrp2*, *Cst3*, *Sfrp1*, *Itm2a*, *Sparc*, and *Ctgf*) and enrichment pathways (Figure 4C,F). This may suggest that vascular smooth muscle development of embryonic OFT was impaired. The above findings suggested that CM I was a group of transitional cells along the trajectory for CM-to-VSMC trans-differentiation. After *PDGFRα* and *PDGFRβ* knockout, CM trans-differentiation into VSMCs may be disturbed and stagnated in a transitional cell state (CM I), which may lead to the failure of OFT remodeling and muscular septum formation. We conducted an in vivo immunofluorescence verification experiment using CM marker cTnT, VSMC marker α-SMA, and CM I cluster-specific marker Sema3c. The result indicated the existence of a CM I cluster, which co-expressed cTnT, α-SMA, and Sema3c and was mainly distributed on the pulmonary artery (Pa) side. In addition, compared with the control group, the distribution area of CM I in the DKO group seemed to expand larger, which was consistent with our single-cell analysis result that CM-to-VSMC differentiation stagnated in the transitional CM I-like state (Figure 4H). However, the CM-to-VSMC differentiation process during OFT separation still remains controversial and only supported by a few recent studies [37]. More research evidence is needed to confirm the existence of this transition process.

Although our current evidence is not enough to clearly explain the potential regulation mechanism, we still make reasonable speculation about the possible effect of *PDGF* knockout on CMs. Through CellChat analysis [26], we noticed that CM I cluster played the main role of the sender in the *PDGF* signaling pathway network, and the target receiver was MS I. The loss of *PDGF* signaling in MS I due to the crest deletion of *PDGFR* might regulate the transformation of more CMs into CM I cells through intercellular feedback communication (Appendix A).

### 3.5. Absence of a Key MS Cluster Associated with Septal Bridge Formation after Pdgfrα and Pdgfrβ Knockout

Next, we concentrated on the two mesenchymal clusters (Figure 5A). Compared with the control group, there was a complete lack of MS II in the DKO group (Figure 5A,B). Analysis of differentially expressed genes between two MS subclusters showed that MS-related genes *Hapln1* and *Twist1* were highly expressed in MS I, with enriched pathways including regulation of epithelial cell proliferation and translation. *Twist1* was previously reported to function in post-migratory CNCC-derived MSs to repress pro-neural factors, thus regulating cell fate determination between ectodermal and mesodermal lineages [49]. The absence of Twist1 resulted in aberrant cardiac neural crest morphogenesis and OFT septation [50]. By contrast, VSMC marker genes *Cxcl12* and *Acta2* were higher in MS II, and enriched pathways included tissue morphogenesis, angiogenesis, and muscle tissue development (Figure 5C,D and Appendix A). Therefore, we postulated that MS II was much closer to the VSMC state than MS I (Figure 5C,D). MS II may be a group of transitional cells along the trajectory of MS-to-VSMC differentiation [37].

Due to the absence of the MS II cluster in the DKO group, only gene expression and enriched pathways of MS I could be compared. Compared with the control group, down-regulated genes of the MS I cluster included *Ptn*, *Tsix,* and *Sfrp2*, and up-regulated genes included *Dbi*, *Ramp2*, and *Nme2* (Figure 5E and Appendix A). Pathway enrichment analysis revealed that down-regulated genes of MS I primarily participated in muscle development, respiratory chain, and oxidative phosphorylation, and up-regulated genes participated in translation and regulation of cellular amide metabolic process (Figure 5F). This indicated that vascular smooth muscle development was impaired in the DKO group.

Previous studies revealed that *Sema3c* was highly expressed in the CM wall and septum of OFT [9,17,51], and *Penk* was specifically expressed in the septum [37]. They were both highly expressed in septal-bridge cells, a group of cells located in the center of the OFT septum and involved in septum formation, according to our data. Of note, our analysis revealed that the expression level of *Sema3c* and *Penk* was higher in MS II (Figure 5G), indicating that MS II represented septal-bridge cells. The MSs in OFT have two main origins: EndoMT and CNCC derivation. We hypothesized that MS II derived from CNCCs and conducted an in vivo immunofluorescence verification experiment. The result showed that all the *Penk^+^* MS II cells had red fluorescence used for CNCC lineage-tracing, which verified our hypothesis (Appendix A). Consistent with our findings, a previous study also reported that the *Penk^+^* cells were mainly localized in the center of OFT, where the aortopulmonary septum formed, and all the *Penk^+^* cells in the heart were derived from CNCCs [29]. When we projected our single-cell data onto previous single-cell data sets of CNCCs [29], we found that *Penk^+^* septal bridge cells from our OFT cell and previous CNCC datasets could mix well into a group and disappeared in DKO mice (Appendix A). Consequently, RNA velocity analysis of all OFT cells (Appendix A) showed that there were two main differentiation trends into VSMCs from CMs and MSs, respectively. Details in Figure 5H revealed that the MS II cluster was a transitional cell state in MS-to-VMSC differentiation. This may indicate that the deletion of MS II in the DKO group is one of the key reasons leading to OFT separation disorder and PTA phenotype. The deletion of *PDGFRα* and *PDGFRβ* caused CNCCs to fail to differentiate into septal-bridge cells.

We performed an RNA Scope experiment in situ hybridization to verify the findings suggested by single-cell transcriptomic sequencing. As shown in Figure 5G, *Sema3c* and *Penk* were co-stained in the center of the OFT septum, the location of septal-bridge cells in the control group. However, there were few signals of *Sema3c* and *Penk* in the DKO group (Figure 5I). These results are consistent with our findings that the MS II cluster with high expression of *Sema3c* and *Penk* was absent after *PDGFRα* and *PDGFRβ* knockout. Altogether, our data found intermediate cell subpopulations involved in either MS-to-VSMC transition or CM-to-VSMC trans-differentiation, which were affected or absent after *PDGFRα* and *PDGFRβ* conditional knockout in CNCCs. This may be the critical mechanism of OFT separation disorder and PTA phenotype.

### 3.6. More VSMC in an Immature State after PDGFRα and PDGFRβ Knockout

As explained above, the cluster proportion and gene expression of CMs and MSs exhibited noticeable changes after *PDGFRα* and *PDGFRβ* knockout. We wondered whether VSMCs demonstrated similar changes in the DKO group (Figure 6A–C). DEGs analysis between two VSMC subclusters showed that the top5 marker genes in VSMC I were *Col1a2*, *Fbn1*, *Eln*, *Rgs5*, and *Myh10*, highly correlated with mature VSMC markers; while genes abundantly expressed in VSMC II were *Tmsb10*, *Tmsb4x*, *Actb*, *H2afz*, and *Mif,* ones with relatively higher expression in fetal MSs (Figure 6D,E and Appendix A). Pathway enrichment analysis showed that the highly expressed genes of VSMC I were closely related to molecules associated with elastic fibers, and those of VSMC II were related to actin cytoskeleton organization (Figure 6F). Combing DEG and pathway enrichment analysis results, we inferred that VSMC I was a relatively mature VSMC cluster. Significantly, the proportion of VSMC II increased in the DKO group, indicating that more VSMCs were relatively immature (Figure 6C).

Compared with the control group, pathway enrichment analysis revealed that down-regulated genes of VSMC I in the DKO group (*Pclaf*, *Tsix*, *Acta2*, *Rrm2*, and *Ranbp1*) primarily participated in the deoxyribonucleotide biosynthetic process and regulation of mRNA splicing via spliceosome. Meanwhile, down-regulated genes of VSMC II in the DKO group primarily participated in the metabolism of RNA, energy metabolism, and mitochondrial biogenesis (Figure 6G,H and Appendix A). In combination with cell proportion, gene expression, and enrichment pathway characteristics, RNA synthesis and splicing were affected after *PDGFRα* and *PDGFRβ* knockout, leading to a higher proportion of VSMCs in a relatively immature manner VSMC II-like state and OFT septum remodeling disorder.

### 3.7. Ligand-Receptor Interaction Changes among MS, CM and VSMC Clusters after PDGFRα and PDGFRβ Knockout

In order to further reveal the changes in the molecular regulation mechanisms related to cell differentiation and function in OFT after *PDGFR* conditional knockout. We identified a total of 77 ligand-receptor (L-R) interactions through CellChat [26] among MS, CM, and VSMC clusters, the involved signaling pathways including integrin, non-canonical Wnt, Tgfβ, Fgf, Pdgf, Notch, Bmp, Ras/MAPK, PI3K/Akt pathway. The control group had 15 specific L-R interactions, and the DKO group had 14 specific L-R interactions (Appendix A).

Of the 77 L-R interactions identified, 20 involved integrins. Integrins can provide crucial adhesive and signaling functions through interactions with the extracellular matrix (ECM) like laminin, fibronectin, and collagen, which are important for paracrine signaling, ECM homeostasis, and the intercellular interactions that impact cardiac cell biology and pathophysiological adaptation in disease [52].

Of non-integrins, we noticed that some L-R interaction changes were mainly related to CMs. Firstly, Bmp signaling L-R interactions [Bmp7-(Bmpr1a + Bmpr2), Bmp7-(Acvr1 + Bmpr2), Bmp4-(Bmpr1a + Bmpr2) and Bmp2-(Bmpr1a + Bmpr2)] to CM I were absent in the DKO group. Bmp is a member of the TGF-β superfamily and plays an important role in embryonic heart development. A previous study reported that Bmp signaling could induce SHF-derived CM differentiation and beating in embryonic mice [53]. It was also reported that Bmp regulated CM proliferation along with Notch signaling [54,55]. Consistent with this, the Notch-relevant L-R interactions (Jag1-Notch2 and Dlk1-Notch2) decreased significantly in the DKO group. Notch signaling was also reported to regulate the differentiation from CM to VSMC [37]. In addition, we observed that Agrn-Dag1 interaction from MS I and VSMC I to CM I and II only existed in the DKO group. Agrin (*Agrn* coding protein) is a component of neonatal ECM, which can attenuate CM maturation and promote CM proliferation [56]. The cooperation effect of Bmp, Notch, and Agrn signaling to CM might lead to the increased proportion of relatively immature CM I-like cells in OFT. Finally, Hspg2-Dag1 L-R interaction was absent in the DKO group. Perlecan (*Hspg2* coding protein) modulates CM activity and is essential for normal tissue development and function [57]. Its loss can lead to a high frequency of OFT malformations, such as the complete transposition of great arteries (TGA) [58]. We noticed that F11r-F11r was only relevant to CM I and II and absent in the DKO group, which might represent self-regulation in CMs.

There are also several L-R interaction changes mainly related to MSs and VSMCs. Ackr3 (originally named RDC1 and CXCR7) related L-R interactions (Mif-Ackr3 and Cxcl12-Ackr3) to MSs were absent in the DKO group, indicating that Ackr3 was mainly expressed in MSs and lost in the DKO group. Cxcl12-Ackr3 heteromeric complexes had been demonstrated to be critical to septum and valve morphogenesis during heart development [59,60]. In contrast, Sdc2-related L-R interactions (Ptn-Sdc2 and Mdk-Sdc2) to MSs were increased in the DKO group. Sdc2 can physically interact with members of the Fgf, Vegf, and Tgfβ superfamilies [61,62]. In cell culture, Sdc2 has been implicated in microvascular angiogenesis through the regulation of cell adhesion and cell migration [63]. In addition, the Mdk-Lrp1 expression pattern showed that this interaction only acted on VSMC I and was absent in the DKO group. Lrp1 function in CNCCs is required for normal OFT development, with other cell lineages along the CNC migratory path playing a supporting role [64].

In summary, the identified L-R interaction changes between the control and DKO group revealed fundamental aspects of molecular regulation mechanism in normal and after PDGFR conditional knockout. These findings also provided clues for further research on the regulatory signal network related to OFT separation.

## 4. Discussion

It is well established that CHDs are a kind of congenital disease with complex etiologies related to genetic and environmental factors. Among them, PTA is a rare but serious developmental conotruncal malformation. This defect is characterized by a single vessel and common valve that originates from the base of the heart supplying the systemic, coronary, and pulmonary circulation. Previous studies have reported that CNCCs in the caudal pharyngeal arches will migrate into the cardiac OFT to form condensed mesenchyme at the junction of the bilateral OFT cushion. This condensed mesenchyme, termed the aorticopulmonary septation complex, divides the OFT into the aorta and pulmonary trunk [65,66]. Disturbance of CNCCs in heart development due to genetic and environmental factors may lead to PTA, but the specific mechanism remains unclear.

Our analysis of the cellular and molecular mechanism of the OFT separation disorder after *PDGFRα* and *PDGFRβ* DKO in CNCCs enabled us to propose a model whereby *PDGFR*-dependent cell trans-differentiations of CNCC-derived cells would set the occurrence of septal bridge formation and OFT septation. First, we constructed a PTA mouse model with a simultaneous knockout of *PDGFRα* and *PDGFRβ* in *Pax3+* NCCs. The analogy between the phenotype of *PDGFRα* and *PDGFRβ* DKO mice and patients suffering from PTA condition enabled us to further explore the cellular and molecular etiology of this complex CHD. Next, we described the changes in cell proliferation, apoptosis, migration, and polarity in the DKO group at the cellular level, verifying and supplementing the findings reported in previous studies [21,36]. Finally, there is still a lack of molecular-level research on malformation caused by *PDGFRα* and *PDGFRβ* knockout. Our single-cell sequencing study provided primary insights into cell subsets and regulatory molecules in OFT septation.

It has been described that disruption of *PDGFRα* signaling can cause defects in CNCC populations, and *PDGFRα* null, and Patch (Ph) mutant allele mice (a naturally occurring *PDGFRα* deletion) indicated that cell survival or matrix deposition, but not apoptosis and proliferation, was the primary function of *PDGFRα* signaling [67,68]. Another study reported that, whereas *PDGFRβ* is essential in certain cell types during embryonic development, its broader role may be masked because of compensation by *PDGFRα* [69]. When combining *PDGFRα* and *PDGFRβ* specifical knockout in *Wnt1*^+^ CNCCs, *Richarte* et al. found a PTA phenotype with 100% penetrance and cellular changes after knockout [21]. In our research, we found no significant difference between DKO and normal embryos concerning CNCC migration, cell proliferation, and apoptosis. Inspired by the study of Darrigrand et al. that used cell polarity to reveal the cytological mechanisms in OFT septation [13], polarity detection of OFT cells taking advantage of *GM130* showed that the concentration trend in cushion merge direction was disturbed after *PDGFRα* and *PDGFRβ* DKO.

In recent years, multiple single-cell RNA-sequencing studies have focused on the field of heart development and CHDs [13,29,37,70]. In OFT development, changes in cell subpopulation and lineage differentiation are key to uncovering the mechanisms underlying normal development and abnormalities [29,37,70]. As a newly-developing technique, single-cell transcriptomics was worthwhile revisiting the model so that it is possible to separate the cell-autonomous from non-cell-autonomous mechanisms. Two single-cell studies recently published by the Zhou group provided profiles of embryonic cardiac OFT development and involvement of NCCs in heart development, respectively [29,37]. Our research further explored the abnormalities of NCCs and aorticopulmonary septum formation during the process of PTA.

We performed single-cell transcriptomic sequencing of 11,998 mouse OFT cells from developmental stages corresponding to OFT septation (E12.5). The large-scale single-cell data empowered us to unbiasedly and systematically dissect the cellular diversity and heterogeneity during OFT development. We identified 10 cell clusters that could be assigned to 6 cell lineages. We provided molecular signatures for the cell lineages and clusters and identified intermediate cell subpopulations (CM I and MS II), which were likely to be involved in either CM to VSMC trans-differentiation or MS to VSMC differentiation after *PDGFRα* and *PDGFRβ* DKO. Particularly, the *Penk^+^* cluster MS II, a relatively small MS subpopulation undergoing MS to VSMC transition, was explicitly associated with the fusion of the OFT cushions and was absent in DKO OFT tissues.

The CM to arterial phenotypic change in OFT has been described in a recent single-cell study on OFT development [37,71,72], demonstrating that trans-differentiation of CMs to arterial components might occur during OFT development in embryonic hearts, yet this view still remains controversial. Consistent with this, our single-cell dataset supports this view of CM to VSMC trans-differentiation by identifying cell clusters representing a continuum of cell state transitions. The expression profile comparison analysis demonstrated that CM cluster I could express CM and VSMC markers simultaneously and might represent an intermediate cell state along the trajectory of CM-to-VSMC trans-differentiation. In vivo immunofluorescence verification experiment also proved that there was a group of cells distributed on the pulmonary artery (Pa) side co-expressing CM and VSMC markers in the early development stage. The expression of SMA-mCherry in the embryo and adult mouse organs reported by Armstrong et al. indicated that α-SMA, which is used as a marker of VSMCs, could be expressed in the embryonic heart and lost in adults [73]. This appeared to provide an explanation for the expression of some smooth muscle markers in the embryonic myocardium. Although our study results might suggest the existence of a group of cells in the intermediate state between CMs and VSMCs, further research is needed to confirm this finding. Meanwhile, we observed that cell trans-differentiation seemed to have stagnated in an intermediate state of CM I in the DKO group. Combined with pathway enrichment analysis results, this may indicate that vascular smooth muscle development of embryonic OFT was impaired after *PDGFRα* and *PDGFRβ* DKO.

In a previous study, a small MS subpopulation involved in the MS differentiation into VSMC was reported by Liu et al. [37]. In our study, we provided further evidence for the existence of the MS-to-VSMC transition and an intermediate MS subpopulation (MS II). Corresponding to previous studies, MS II highly expressed intermediate MS subpopulation marker *Penk* and *Sema3c*. Furthermore, this cluster was mainly condensed into a compact septal-bridge structure, as shown in Figure 5I. All these observations demonstrated the role of cluster MS II in OFT separation formation. Importantly, we found that cluster MS II was absent after *PDGFRα* and *PDGFRβ* DKO, which indicated the disturbance of MS differentiation into VSMC and failure of cell condensation into a septal bridge, providing mechanistic insights into OFT septation disorder.

Our single-cell sequencing data analysis of CMs, MSs, and VSMCs revealed that VSMCs could be downstream of the differentiation of both CMs and MSs. First, as two crucial differentiation sources of VSMCs, CMs and MSs, especially the two transitional cell subpopulations CM I and MS II identified in this study, had undergone important changes after *PDGFRα* and *PDGFRβ* knockout. Among them, CM differentiation into VSMCs stagnated in a *Sema3c*^+^ CM I state, while the MS II cluster expressing *Penk* and *Sema3c* disappeared in the DKO group. Besides, VSMCs also changed gene expression with a more immature transcriptional state.

Cell polarity may reflect the condensation of OFT cushion cells, which is an important segment in aortopulmonary septum formation. Our analysis results indicated that the cell polarity disorder in the DKO group might be caused by the down-regulation of Sema3c (a key molecule regulating the directional migration of cushion cells) expression in specific cell subpopulations. In addition, differentiation disorders of MSs into VSMCs may also make cushion cells acquire a less polarized morphology. These changes together caused the abnormal condensation of OFT cushion cells, which led to the occurrence of PTA.

There are still some limitations to this research. Using single-cell RNA-sequencing, our study provided new insights into the OFT septation disorder in PTA formation. However, the molecular mechanisms and downstream regulatory networks still need to be further explored and elucidated. In addition, more embryo samples at different developmental stages are needed to reconstruct this developmental process more comprehensively.

## Figures and Tables

**Figure 1 genes-13-01708-f001:**
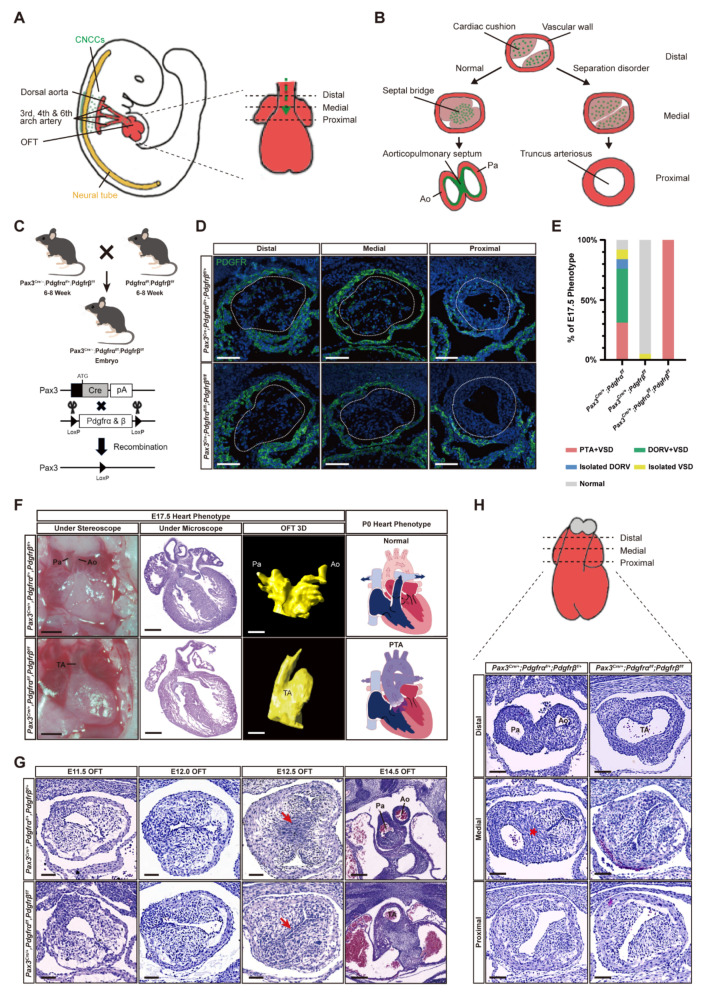
Animal model establishment and phenotypic identification. (**A**) Schematic representation showing that CNCCs (orange) have reached the heart region (red) in the E10.5 mouse embryo and then migrated through the distal-proximal axis of the OFT. (**B**) The schematic representation of OFT transverse sections shows that CNCC condensation and OFT separation occurred along the OFT distal-proximal axis, while these events disappeared in the separation disorder situation. (**C**) Target *PDGFRα* and *PDGFRβ*-conditional-knockout (*Pax3^cre/+^; PDGFRα^fl/fl^; PDGFRβ^fl/fl^*) embryos were obtained by hybridization between *Pax3^cre/+^; PDGFRα^fl/+^; PDGFRβ^fl/fl^* and *PDGFRα^fl/fl^; PDGFRβ^fl/fl^* mice. *Pax3-Cre* was used to knockout *PDGFRα* and *PDGFRβ* in CNCCs specifically. (**D**) The efficiency of *PDGFRα* and *PDGFRβ* knockout in CNCCs by immunofluorescence shows that the expression of *PDGFRα* and *PDGFRβ* was lower in the OFT cushion region in the DKO group. *n* = 3 vs. 3. Scale bar, 50 µm. (**E**) Phenotypic statistics of *PDGFRα/β* single and DKO group at E17.5. *n* = 15 vs. 48. (**F**) The phenotypic confirmation at E17.5 with multiple methods (stereoscope, microscope, and 3D OFT reconstruction) showed that the embryos in the DKO group had a PTA phenotype. *n* = 15 vs. 48. Scale bar, 300 µm. (**G**) Timepoint confirmation of OFT separation. The cross-section of embryonic OFT showed that OFT separation had not occurred until E12.5. At E12.5, OFT started separation in the control group, while septum was not formed in the DKO group (red arrow). E11.5, *n* = 8 vs. 8; E12.0, *n* = 10 vs. 8; E12.5, *n* = 12 vs. 9; E14.5, *n* = 7 vs. 3. Scale bar, 50 µm. (**H**) The phenotypic confirmation on transverse sections across the OFT at three distinct distal-proximal levels in E12.5 embryos with the indicated genotype. *n* = 12 vs. 9. Scale bar, 50 µm. Pha, pharyngeal arch; V, ventricle; Pa, pulmonary trunk; Ao, aorta; TA, truncus arteriosus; OFT, outflow tract; DKO, double knockout; PTA, persistent truncus arteriosus; CNCCs, cardiac neural crest cells; *, septal bridge.

**Figure 2 genes-13-01708-f002:**
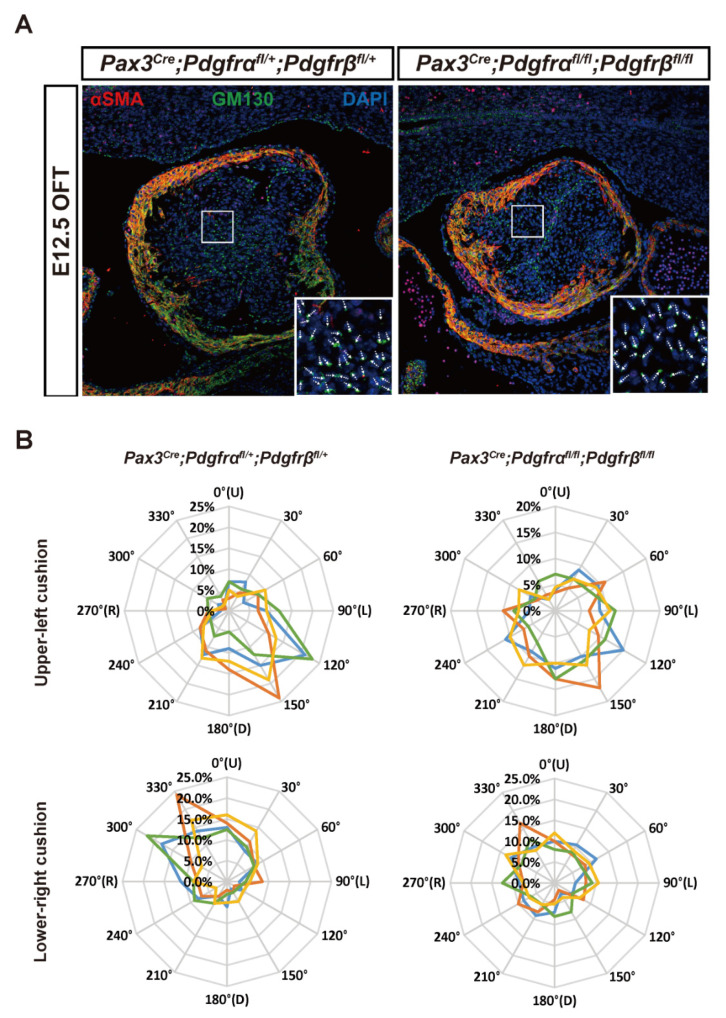
OFT cell polarity difference between DKO and control group. (**A**) The cellular polarity of OFT cells was examined with anti-GM130 (a marker for Golgi) and DAPI (a marker for nucleus). *n* = 4 vs. 4. Scale bar, 50 µm. (**B**) A clockwise angle between the vector from the center of the nucleus to the Golgi apparatus and the right-left axis (0–180°) was measured in individual OFT cushion cells. The percentage of cell numbers in each angle range was plotted on radar graphs, which showed a similar distribution pattern of data obtained from 4 embryos in each graph (marked with different colors). In the upper-left and lower-right cushion of the control group, the vector angle was frequently distributed at 120–150° and 300–330° (*p* < 0.01, v2 test), respectively, whereas the polarity of the vector angle was not significant in DKO group (*p* = 0.13, v2 test). OFT, outflow tract; DKO, double knockout.

**Figure 3 genes-13-01708-f003:**
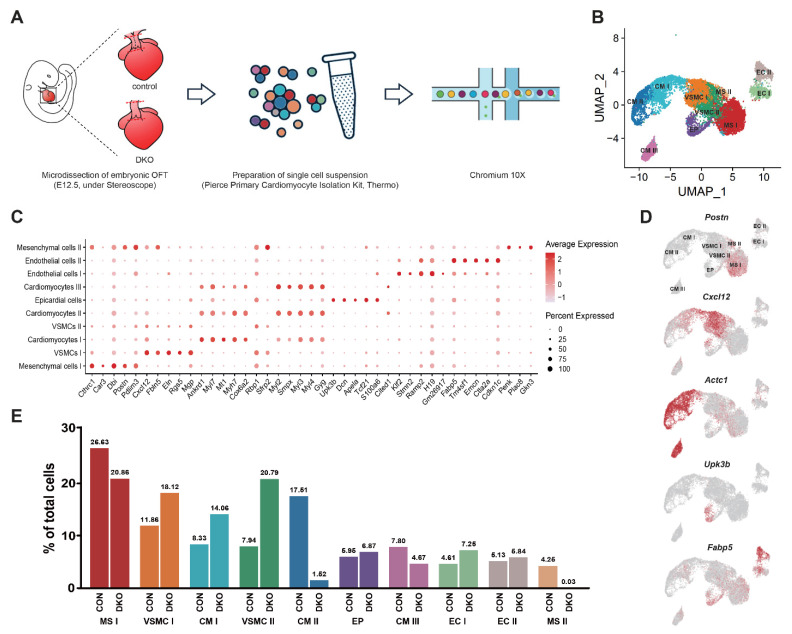
Single-cell transcriptome sequencing revealed a profile of cell cluster, lineage, and proportion. (**A**) Overview of the experimental procedure. Embryonic OFT was micro-dissected (red dotted line) under a stereoscope and built library by Chromium 10× for single-cell transcriptome sequencing. (**B**) Unsupervised clustering of aggregate OFT cells revealed 10 cell clusters projected on UMAP plots. (**C**) Dot plot showing expression of top up-regulated genes across OFT cell clusters. (**D**) Expression of select marker genes across OFT cells as visualized on UMAP plots (MSs: *Postn*; VSMCs: *Cxcl12*; CMs: *Actc1*; epicardial cells: *Upk3b*; and endothelial cells: *Fabp5*). (**E**) Cell population percentages of each cluster between DKO and control group. MS, mesenchymal cell; CM, cardiomyocyte; VSMC, vascular smooth muscle cell; EP, epicardial cell; EC: endothelial cell; CON, control; DKO, double knockout; UMAP, uniform manifold approximation and projection; OFT, outflow tract.

**Figure 4 genes-13-01708-f004:**
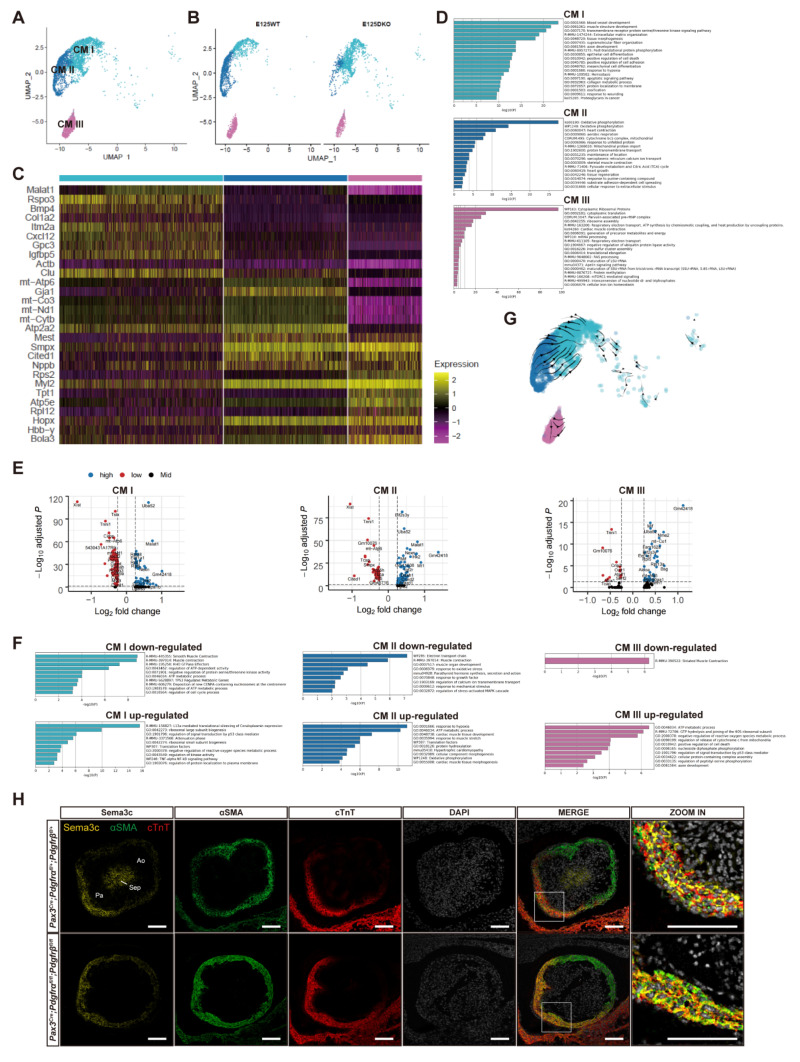
CM clusters and changes after *PDGFRα* and *PDGFRβ* knockout. (**A**) UMAP plot of all captured CM populations colored by cluster identity (CM I, II, and III). (**B**) Cell cluster distribution of the DKO and control group on UMAP plots revealed that there was a significant difference in the proportion of CM I and II between the DKO and control group. (**C**) Heatmap showing top 10 cluster-specific genes for each CM subcluster. (**D**) Functional enrichment of genes significantly expressed in CM I, II, and III. (**E**) DEGs of CM I, II, and III between DKO and control group. (**F**) Functional enrichment of genes significantly up-regulated and down-regulated in CM I, II, and III between the DKO and control group. (**G**) Direction and rate of cellular state changes inferred by RNA velocity analysis between CM clusters. (**H**) In vivo immunofluorescence verification experiment using CM marker cTnT, VSMC marker α-SMA, and CM I cluster specific marker Sema3c showed that CM I existed and was mainly distributed on the pulmonary artery side. *n* = 5 vs. 5. Scale bar, 50 µm. UMAP, uniform manifold approximation and projection; DEG, differentially expressed genes; CON, control; DKO, double knockout; MS, mesenchymal cell; CM, cardiomyocyte; VSMC, vascular smooth muscle cell; EP, epicardial cell; EC: endothelial cell; OFT, outflow tract; Pa, pulmonary trunk; Ao, aorta; Sep, septal bridge.

**Figure 5 genes-13-01708-f005:**
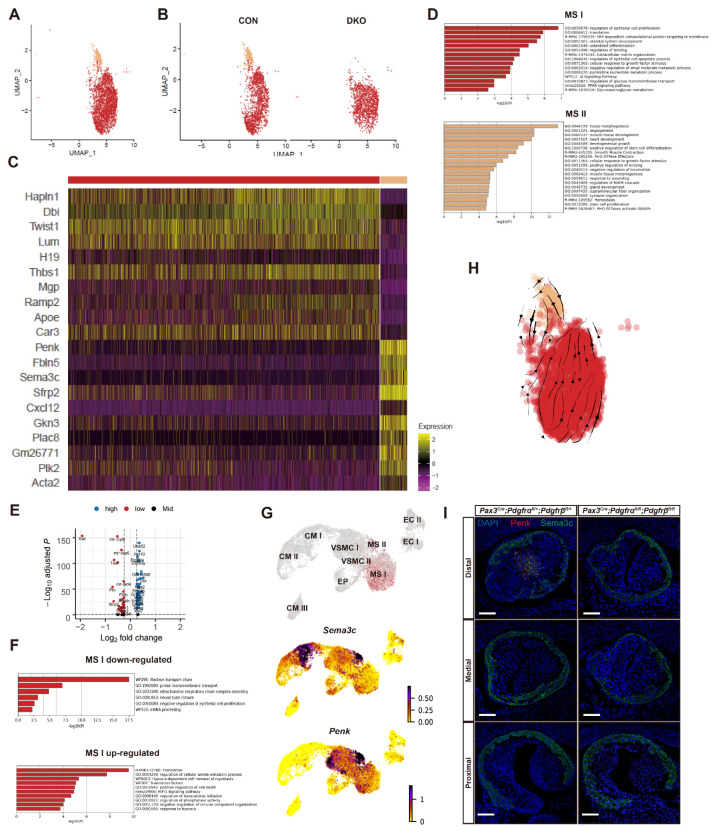
MS clusters and changes after *PDGFRα* and *PDGFRβ* knockout. (**A**) UMAP plot of all captured MS populations colored by cluster identity (MS I and II). (**B**) Cell cluster distribution of the DKO and control group on UMAP plots revealed that cluster MS II was absent in the DKO group. (**C**) Heatmap showing DEGs between MS I and II. (**D**) Functional enrichment of genes significantly up-regulated in MS I and II. (**E**) DEGs of MS I after *PDGFRα* and *PDGFRβ* DKO. (**F**) Functional enrichment of genes significantly up-regulated and down-regulated in MS I between the DKO and control group. (**G**) *Sema3c* and *Penk* were both highly expressed on MS II, indicating that MS II represented the MSs of the septal bridge in OFT. (**H**) Direction and rate of cellular state changes inferred by RNA velocity analysis. (**I**) *Penk^+^* and *Sema3c^+^* MS II cluster cell absence in the middle of OFT cushions after *PDGFRα* and *PDGFRβ* knockout was confirmed by single-molecule fluorescent in situ hybridization. *n* = 3 vs. 3. Scale bar, 50 µm. UMAP, uniform manifold approximation and projection; DEG, differentially expressed genes; CON, control; DKO, double knockout; MS, mesenchymal cell; OFT, outflow tract.

**Figure 6 genes-13-01708-f006:**
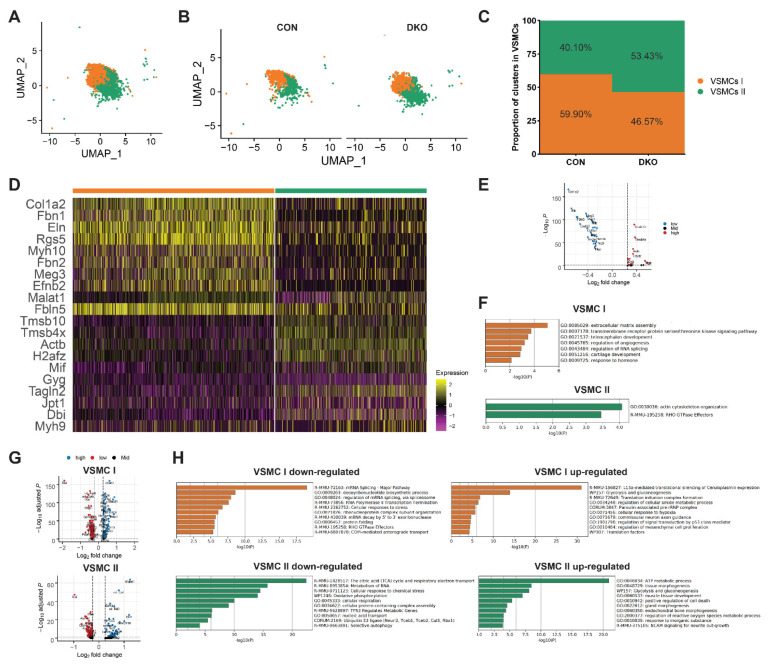
VSMC changes after *PDGFRα* and *PDGFRβ* knockout. (**A**) UMAP plot of all captured VSMC populations colored by cluster identity (VSMC I and II). (**B**) Cell cluster distribution of the DKO and control group on UMAP plots. (**C**) The proportion of VSMC I and II cells between the DKO and control group. (**D**) Heatmap showing DEGs among VSMC subgroups. (**E**) Expression of marker genes across VSMC clusters as visualized on UMAP plots (VSMC I: *Col1a2* and *Fbn1*; VSMC II: *Tmsb10* and *Tmsb4x*). (**F**) Functional enrichment of genes significantly up-regulated in VSMC I and II. (**G**) DEGs of VSMC I and II after *PDGFRα* and *PDGFRβ* DKO. (**H**) Functional enrichment of genes significantly up-regulated and down-regulated in VSMC I and II between the DKO and control group. UMAP, uniform manifold approximation and projection; DEG, differentially expressed genes; CON, control; DKO, double knockout; VSMC, vascular smooth muscle cell.

## Data Availability

The original contributions presented in the study are included in the article/Appendix A. Further inquiries can be directed to the corresponding authors.

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
