# Peer review of "Single Cell Sequencing Reveals Mechanisms of Persistent Truncus Arteriosus Formation after PDGFRα and PDGFRβ Double Knockout in Cardiac Neural Crest Cells"

_genes, 2022, doi:10.3390/genes13101708_

Round 1

Author Response

Dear reviewer:

Thanks very much for taking your time to review this manuscript. We have responded to the comments from reviewers and editors point by point in red font. Please see the attachment for details.

Best wishes!

Reviewer 2 Report

The report describes the authors established a persistent truncus arteriosus (PTA) mouse model using double knockout (DKO) of PDGFRα and PDGFRβ in Pax3+ cardiac neural crest cells (CNCCs) and showed that PDGFRα and PDGFRβ are involved in cell polarity of CNCC-derived cells and are important for the formation of outflow tract (OFT) septation. In addition, single-cell RNA sequencing revealed that vascular smooth muscle cells (VSMCs) from cardiomyocytes (CMs) and mesenchymal cells (MSs) respectively were interrupted and a Sema3 and Penk+ transitional MS II cluster closely related to cell condensation into OFT septum disappeared in the DKO models. The presented findings are of interest giving new insights into a possible pathogenic mechanism of PTA patients. However, the text is redundant, and some figures are unclear. The following major issues need the authors' attention.

 Major Comments:

PDGFRα and PDGFRβ were shown to be involved in cell polarity in CNCC-derived cells. In the single-cell RNA sequencing, what were the differences in the expression patterns of molecules involved in polarity formation between CON and DKO? What is the relationship between cell polarity and the process of differentiation from CMs or MSs to VSMCs? It would be more interesting to discuss the interpretation of RNA sequencing results focusing on the molecular mechanisms of polarity formation.

Page4 line95-101  I am wondering why DKO was used instead of PDGFRαKO, even though previous reports have shown no abnormalities in PDGFRβKO. It would be better to describe the advantages of using DKO instead of PDGFRαKO.

Page9 line 322-324   In this paragraph, time-course changes of OFT tissue (Figure 1G) are also important information, which is linked to further experiments using E12.5 tissue. Therefore, it is recommended to include that information in this summary text.

Figure 1C, D  Quantitative confirmation of protein or mRNA expression of PDGFRα and PDGFRβ in the OFT region of DKO mice would be needed to rule out the possibility of low efficiency of suppression by the Cre/loxP system. Representative images are not fully convincing.

Figure 4H, 5I  Representative images are not fully convincing. Please indicate from how many individual experiments the images were obtained. It is better to evaluate quantitatively if possible.

Figure 1D  It is unclear what Green indicates. It would be useful to determine the localization of PDGFRα and PDGFRβ, respectively, then to compare them with the localization of Pax3+ CNCC to confirm the match in localization.

Figure 1D, 2A, 3H, and 5I,   The photos of all fluorescent staining are unclear, so it is difficult to determine if the contents are suitable. Please replace them with clearly visible ones.

Minor Comments

Page1 line38  The reference papers are too old and may differ from the current situation. Please refer to recent papers.

Figure 1F  The phenotyping criteria will need to be clearly indicated in the methods.

Figure2B  To prove the certainty of this result, information on the total number of examined cells is needed.

Page7 line229  Please break line and cancel Bold.

Page8 line275  Figure1BFigure1C (Please correct all of the incorrect alphabets of Figure 1 )

Page9 line 324  (Figure 1A) may be unnecessary.

Page14 line434   (H) should be bolded.

Author Response

(The authors gave the same response as above.)

Reviewer 3 Report

In this study, Chen et al constructed the PTA mouse model with PDGFRα and PDGFRβ double knockout (DKO) in Pax3+ CNCCs. Based on this model, they found the condensation failure in the OFT septum of CNCC-derived cells due to disturbance of cell polarity in the DKO group. Strikingly, they also combined single-cell transcriptome data analysis and featured that cardiomyocytes and mesenchymal cells transition will be disturbed by PDGFRα and PDGFRβ double knockout. Overall, the study is comprehensive and insightful.

1. Line 392 the authors may consider adding the specific methods of differential expression analysis and functional enrichment analysis performed in this article.

2. Line 666-conotrunk? conotruncal?

Author Response

(The authors gave the same response as above.)

Round 2

Reviewer 1 Report

The authors have made a careful and conscientious effort to reply to each point in my earlier review. I find the manuscript much improved as a result.

Language-wise, the new lines 734-736 might be easier to read as "Since single-cell transcriptomics is a new technique developed since earlier work on the role of PDGF receptors in cardiac neural crest cells (Richarte et al. 2007), we considered it worthwhile to revisit the model to distinguish cell-autonomous from non-cell-autonomous mechanisms." Just before that, the authors can remove "Especially" from the previous sentence.

I think the authors could prepare a supplementary figure with the craniofacial phenotype of the 6 embryos with the profound facial hypoplasia shown to be in Figure R2. Why do they think it did not happen in all embryos?

To previous point 5, the authors have replied that they could not photograph their microdissections, and that they consider it likely that they included the valves of the outflow tract. I agree with the latter, but had found that Figure 3A was a little ambiguous on this point. They can leave it as is.

Figure 1 overall is much improved, and I appreciate the authors adding either their primary data for the cell orientation studies or the references to their deposit in publicly accessible repositories for the single-cell data.

Author Response

(The authors gave the same response as above.)

Reviewer 2 Report

The presented manuscript is revised adequately.

Author Response

Thanks very much for taking your time to review this manuscript. We really appreciate all your comments and suggestions. Your comments were highly insightful and enabled us to greatly improve the quality of our manuscript.